# Shank3 modulates sleep and expression of circadian transcription factors

**Ashley M Ingiosi[1†], Hannah Schoch[1†], Taylor Wintler[1], Kristan G Singletary[1], Dario Righelli[2,3], Leandro G Roser[1], Elizabeth Medina[1], Davide Risso[4,5], Marcos G Frank[1]\*, Lucia Peixoto[1]\***

[1]Department of Biomedical Sciences, Elson S. Floyd College of Medicine, Washington State University, Spokane, United States; [2]Istituto per le Applicazioni del Calcolo "M. Picone", Consiglio Nazionale della Ricerche, Napoli, Italy; [3]Dipartimento di Scienze Aziendali Management & Innovation Systems, University of Fusciano, Fisciano, Italy; [4]Department of Statistical Sciences, University of Padova, Padova, Italy; [5]Division of Biostatistics and Epidemiology, Department of Healthcare Policy and Research, Weill Cornell Medicine, New York, United States

**\*For correspondence:**
marcos.frank@wsu.edu (MGF);
lucia.peixoto@wsu.edu (LP)

[†]These authors contributed equally to this work

**Competing interests:** The authors declare that no competing interests exist.

**Abstract** Autism Spectrum Disorder (ASD) is the most prevalent neurodevelopmental disorder in the United States and often co-presents with sleep problems. Sleep problems in ASD predict the severity of ASD core diagnostic symptoms and have a considerable impact on the quality of life of caregivers. Little is known, however, about the underlying molecular mechanisms of sleep problems in ASD. We investigated the role of *Shank3*, a high confidence ASD gene candidate, in sleep architecture and regulation. We show that mice lacking exon 21 of *Shank3* have problems falling asleep even when sleepy. Using RNA-seq we show that sleep deprivation increases the differences in prefrontal cortex gene expression between mutants and wild types, downregulating circadian transcription factors *Per3*, *Bhlhe41*, *Hlf*, *Tef*, and *Nr1d1*. *Shank3* mutants also have trouble regulating wheel-running activity in constant darkness. Overall, our study shows that *Shank3* is an important modulator of sleep and clock gene expression.
DOI: https://doi.org/10.7554/eLife.42819.001

## Introduction

Autism Spectrum Disorder (ASD) is the most prevalent neurodevelopmental disorder in the United States (diagnosed in 1 in 59 children [*Baio et al., 2018*]). The core symptoms of ASD include social and communication deficits, restricted interests, and repetitive behaviors (*American Psychiatric Association, 2013*). In addition, several studies show that individuals with ASD report a variety of co-morbid conditions including sleep problems and altered circadian rhythms (*Glickman, 2010*). It is estimated that 40–80% of the ASD population experience sleep disorders that do not improve with age (*Johnson et al., 2009*). More specifically, people with ASD have problems falling and staying asleep (*Hodge et al., 2014*). A recent study showed that sleep problems co-occur with autistic traits in early childhood and increase over time, suggesting that sleep problems are an essential part of ASD (*Verhoeff et al., 2018*). Indeed, sleep impairments are a strong predictor of the severity of ASD core symptoms as well as aggression and behavioral issues (*Cohen et al., 2014*; *Tudor et al., 2012*). Although a great number of studies documented sleep problems in ASD, little is known about the underlying molecular mechanisms.

To better understand the mechanisms underlying sleep problems in ASD, we need animal models that closely recapitulate sleep phenotypes observed in clinical populations. The study of genetic animal models of ASD, in which a genetic abnormality that is known to be associated with ASD is introduced, has provided valuable insight into the molecular mechanisms underlying ASD (*de la Torre-*

**eLife digest** Autism spectrum disorder (ASD) is the most common neurodevelopmental disorder in the United States. People with ASD tend to have difficulties with communication and social interactions, restricted interests, and may repeat certain behaviors. They also often struggle to fall or stay asleep. Sleep deprivation may exacerbate other symptoms of the disorder. This makes life more difficult for both the person with ASD and their caregivers. Scientists do not yet know what causes sleep difficulties in people with ASD.

Unraveling the complex genetics that underlie ASD may help scientists better understand ASD-related sleep difficulties. One possible genetic culprit for sleep difficulties in ASD is a gene called *SHANK3*. Patients with an ASD-associated condition called Phelan-McDermid syndrome are often missing the *SHANK3* gene. They also often have sleep problems.

Now, Ingiosi, Schoch et al. show that both patients with Phelan-McDermid syndrome and mice with a mutation in the *Shank3* gene have problems falling asleep. Using a registry that collects genetic and sleep information on people with Phelan-McDermid syndrome, Ingiosi, Schoch et al. found that people who are missing *SHANK3* frequently have trouble falling asleep and wake up many times each night. Mice missing part of the *Shank3* gene also had difficulty falling asleep, even after they have been deprived of sleep.

Mice naturally have a daily pattern of sleep and activity. This 24-hour activity cycle is maintained by an internal circadian clock. In mice missing part of *Shank3*, the circadian clock genes are not turned on correctly. These genes were less active in mice missing *Shank3*, and this difference worsened with lack of sleep. These mice also ran less on a wheel than typical mice when kept in total darkness, even though the pattern of activity did not change. The experiments suggest that *Shank3* controls sleep, likely through its effects on circadian clock genes. Learning more about what causes these sleep problems may help scientists develop ways to improve sleep in people with ASD and Phelan-McDermid syndrome.

DOI: https://doi.org/10.7554/eLife.42819.002

*Ubieta et al., 2016*). These models include Fragile X syndrome, 16p11.2 deletion syndrome, cortical dysplasia-focal epilepsy (CDFE) syndrome, and mutations in neuroligins, neurexins, and shank genes among others. However, sleep research in animal models of ASD is limited and has not yet revealed the underlying mechanisms of sleep issues associated with ASD. Studies using a fly model of Fragile X syndrome reported an increase in sleep which is in contrast to what is observed in the clinical population (*Bushey et al., 2009*). The opposite phenotype was reported in a Fragile X mouse model, displaying instead an age-dependent reduction in activity during the light phase (i.e the mouse subjective night) (*Boone et al., 2018*). *Neuroligin 1* knockout mice spend more time asleep (*El Helou et al., 2013*), but mice with mutations in *Neuroligin 2* spend less time asleep and more time awake (*Seok et al., 2018*). Mice with a missense mutation in *Neuroligin 3* show normal sleep behavior (*Liu et al., 2017*), but rats with a deletion mutation in *Neuroligin 3* spend less time in non-rapid eye movement (NREM) sleep than wild type rats (*Thomas et al., 2017*). Mutant rat models of CDFE syndrome show longer waking periods while the mutant mice show fragmented wakefulness (*Thomas et al., 2017*). Mice carrying a 16p11.2 deletion syndrome sleep less than wild type mice, but only males are affected (*Angelakos et al., 2017*). More importantly, issues with sleep onset, the most prominent feature of sleep problems in ASD patients, have not been evaluated in animal models of ASD.

In this study, we examined sleep in Phelan-McDermid syndrome (PMS) patients with *SHANK3* mutations and in a mutant mouse with a deletion in *Shank3* exon 21 (Shank3$^{\Delta C}$). PMS is a syndromic form of ASD characterized by gene deletions affecting the human chromosomal region 22q13.3 (*Phelan and McDermid, 2012*), particularly the neuronal structural gene *SHANK3*. Individuals with PMS have high rates of intellectual disability, difficulties in communication and motor function, and approximately 84% fit the core diagnostic criteria for ASD (*Soorya et al., 2013*). There is also a high rate of sleep problems in PMS (*Bro et al., 2017*). Mice with mutations in *Shank3* recapitulate multiple features of both ASD and PMS (*Bozdagi et al., 2010*; *Dhamne et al., 2017*; *Jaramillo et al., 2017*; *Jaramillo et al., 2016*; *Kouser et al., 2013*; *Peça et al., 2011*; *Speed et al., 2015*), including

cognitive impairment, deficits in social behavior, and impaired motor coordination. We show that PMS patients have trouble falling and staying asleep similar to what is observed in the general ASD population. We also show that Shank3$^{\Delta C}$ mice sleep less than wild type mice when sleep pressure is high, have reduced sleep intensity (using an accepted electroencephalographic (EEG) metric), and have delayed sleep onset following sleep deprivation. To identify molecular mechanisms underlying sleep changes in Shank3$^{\Delta C}$ mice, we carried out genome-wide gene expression studies. We previously showed that genome-wide gene expression analysis is a valuable approach to understand the molecular mechanisms underlying the detrimental effects of sleep deprivation (*Gerstner et al., 2016*; *Vecsey et al., 2012*). In this study, we found that sleep deprivation sharply increases the differences in gene expression between Shank3$^{\Delta C}$ mutants and wild type mice, downregulating circadian transcription factors *Per3*, *Bhlhe41* (*Dec2*), *Hef*, *Tlf*, and *Nr1d1* (*Rev-erbα*). We also show that Shank3$^{\Delta C}$ mice are unable to sustain wheel-running activity in constant darkness. Overall, these studies demonstrate that *Shank3* is an important modulator of sleep that may exert its effect through the regulation of circadian transcription factors. Our findings may lead to a deeper understanding of the molecular mechanisms underlying sleep problems in ASD. This may one day lead to the development of successful treatments or interventions for this debilitating comorbidity.

## Results

### Phelan-McDermid syndrome patients have problems falling and staying asleep

Recent studies suggest that sleep problems may be present in a substantial number of PMS patients and may be an important factor for caregivers' well-being (*Bro et al., 2017*). We obtained genetic and sleep questionnaire data from the Phelan-McDermid Syndrome International Registry (PMSIR) to estimate the frequency and age of onset of sleep problems in PMS individuals carrying a *SHANK3* deletion. In parallel, we surveyed the clinical literature to estimate the prevalence of sleep problems in ASD (*Andersen et al., 2008*; *Cotton and Richdale, 2006*; *Gail Williams et al., 2004*; *Giannotti et al., 2008*; *Giannotti et al., 2006*; *Krakowiak et al., 2008*; *Liu et al., 2006*; *Miano et al., 2007*; *Paavonen et al., 2008*; *Polimeni et al., 2005*; *Richdale and Prior, 1995*; *Tani et al., 2003*; *Thirumalai et al., 2002*; *Wiggs and Stores, 2004*) and typically developing populations (*Anders and Eiben, 1997*; *Baweja et al., 2013*; *Bixler et al., 2009*; *Hysing et al., 2013*; *Leger et al., 2012*; *Loessl et al., 2008*; *Lozoff et al., 1985*; *Lumeng and Chervin, 2008*; *Ohayon et al., 2000*; *Pallesen et al., 2008*; *Patzold et al., 1998*; *Sadeh et al., 2000*). *Figure 1* shows that PMS patients have trouble falling asleep and experience multiple night awakenings starting at about 5 years of age. Those difficulties translate to reduced time asleep particularly during adolescence. Although total sleep time seems to improve in adulthood for PMS patients, that improvement is accompanied by an increase in parasomnias. Problems falling and staying asleep persist regardless of age. The frequency of problems falling and staying asleep in PMS patients is similar to what is observed in the general ASD population and much higher than in typically developing individuals (*Figure 1—source data 1*).

### Shank3$^{\Delta C}$ mice sleep less during the dark phase and show reduced sleep intensity

To determine if Shank3$^{\Delta C}$ mice have deficits in spontaneous sleep, undisturbed baseline EEG and electromyographic (EMG) recordings were obtained from wild type (WT) and Shank3$^{\Delta C}$ mice. There was a significant period x genotype interaction for total sleep time (i.e. total time spent in NREM and rapid eye movement (REM) sleep) during the light (hours 1–12) and dark (hours 13–24) periods ($F_{(1,28)}$ = 5.198, p=0.036). Posthoc pairwise comparisons using Sidak correction showed that Shank3$^{\Delta C}$ mice slept less than WT mice during the dark period (p=0.045; *Table 1*). To determine in which arousal states and hours this effect was most pronounced, we examined hourly time in state data for wakefulness, NREM sleep, and REM sleep. Repeated measures ANOVA over the full 24 hr recording period found significant time x genotype interactions for wakefulness ($F_{(23,345)}$ = 2.419, p<0.0001), NREM sleep ($F_{(23,345)}$ = 2.357, p=0.001), and REM sleep ($F_{(23,345)}$ = 2.175, p=0.002) (*Figure 2A*). Posthoc comparisons found the most pronounced differences at hour 19 when Shank3$^{\Delta C}$ mice spent more time in wakefulness (p<0.0001) and less time NREM (p<0.0001) and REM

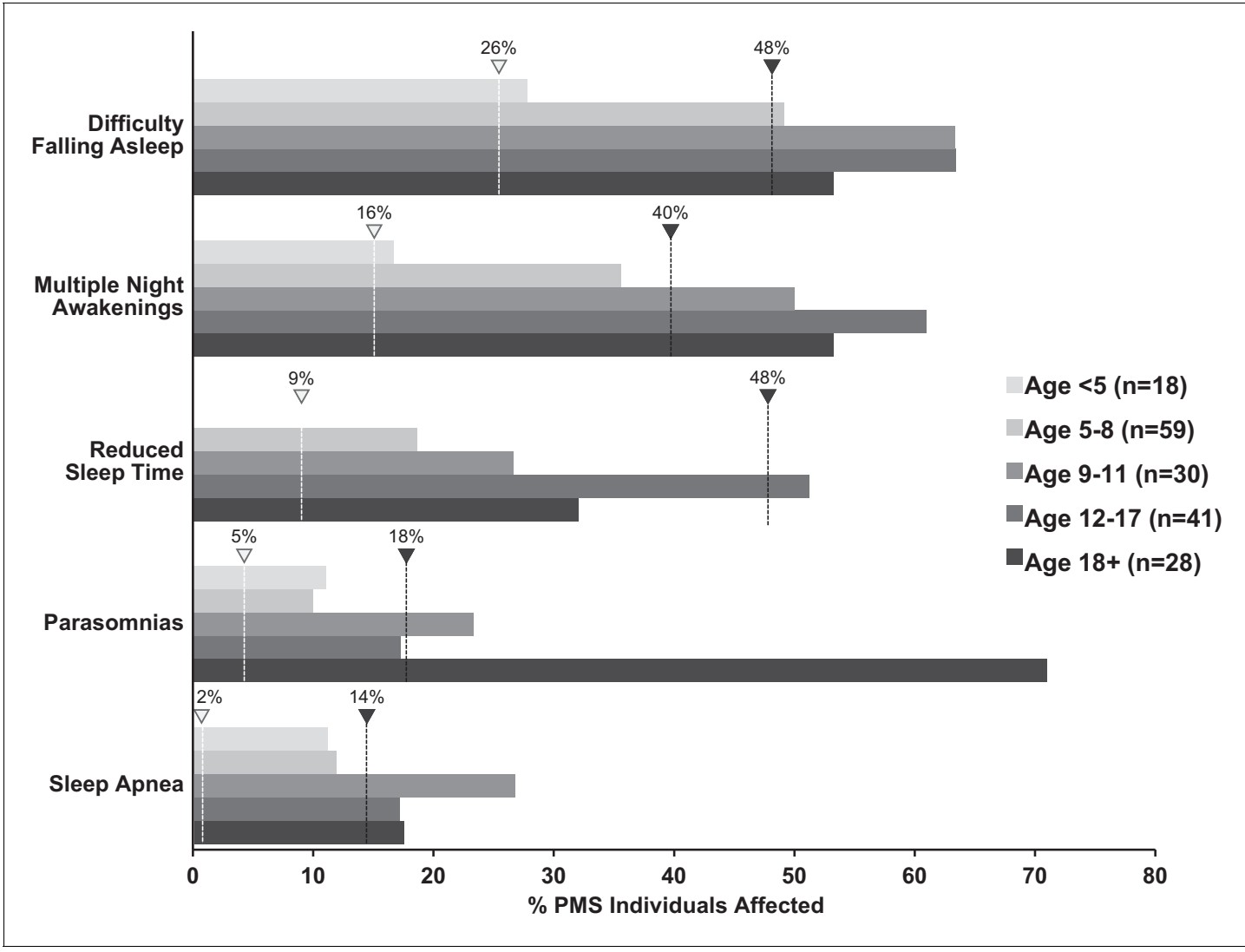

**Figure 1.** Increased incidence of sleep problems reported in individuals with Phelan-McDermid syndrome (PMS) compared to typically developing (TD) individuals. Dashed line indicates median incidence observed in TD (white marker) and ASD (black marker) populations (values from *Figure 1—source data 1*).

DOI: https://doi.org/10.7554/eLife.42819.003

The following source data is available for figure 1:

**Source data 1.** Incidence of sleep problems in Autism Spectrum Disorder (ASD) compared to typically developing (TD) individuals.

DOI: https://doi.org/10.7554/eLife.42819.004

(p=0.040) sleep (*Figure 2A*) marking the beginning of an overall trend for increased wakefulness during the last half of the dark period. Analysis of the sleep architecture found time x genotype

**Table 1.** Shank3$^{\Delta C}$ mice sleep less than wild type mice during the dark period.
Repeated measures ANOVA with posthoc pairwise comparisons using Sidak correction. Values are means ± SEM for wild type (n = 10) and Shank3$^{\Delta C}$ (n = 10). *p<0.05, difference from wild type.

| Total Sleep Time (%) | Wild Type | Shank3$^{\Delta C}$ | p-value |
|---|---|---|---|
| Light Period | 62.90 ± 1.57 | 62.96 ± 1.65 | 0.980 |
| Dark Period | 28.72 ± 2.41 | 21.16 ± 2.54 | **0.045\*** |

DOI: https://doi.org/10.7554/eLife.42819.009

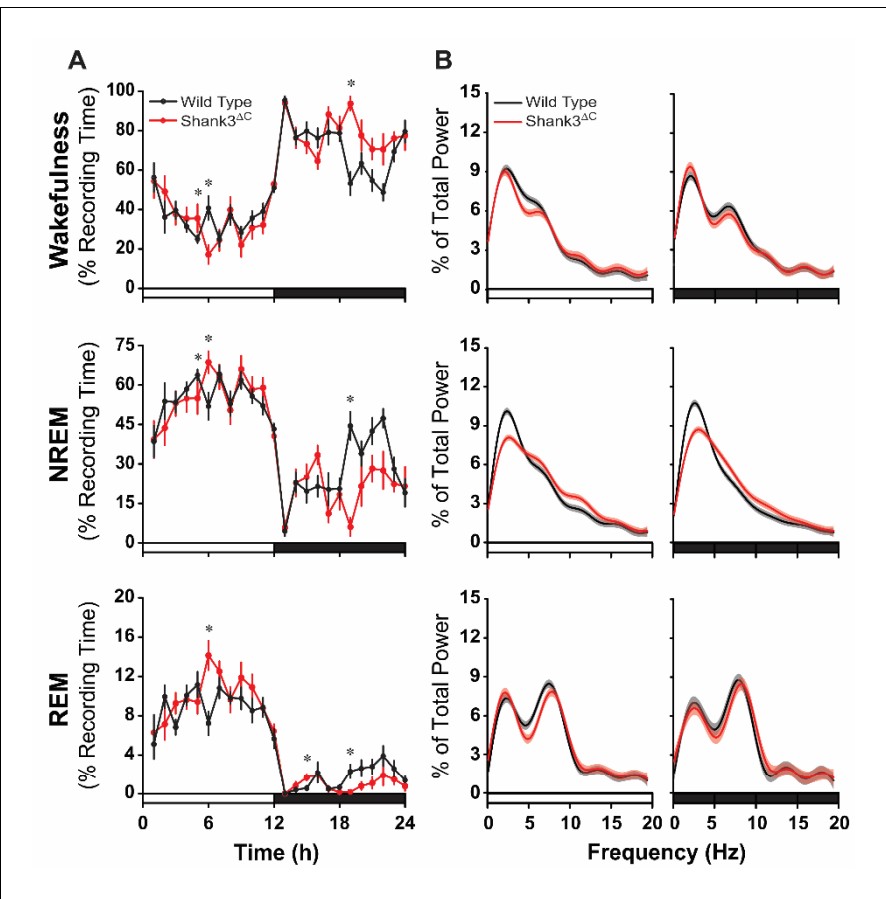

**Figure 2.** Shank3^ΔC mice sleep less during the dark period and show altered EEG spectral power under baseline conditions. The rows represent the vigilance states of wakefulness (top), NREM sleep (middle), and REM sleep (bottom). (**A**) Time spent in wakefulness and sleep shown as percentage of recording time per hour. Values are means ± SEM. Repeated measures ANOVA with posthoc pairwise comparisons using Sidak correction; *p<0.05, difference from wild type. (**B**) EEG spectral power normalized as a percentage of the total state-specific EEG power for the light period (left) and dark period (right) fit to smooth curves (solid lines) and expressed with 95% confidence intervals (gray and red shading). Non-overlap of the 95% confidence interval denotes statistically significant differences. The open bars on the x-axis denote the light period and the filled bars denote the dark period of the light:dark cycle. Wild type (n = 10) and Shank3^ΔC (n = 10) mice.

DOI: https://doi.org/10.7554/eLife.42819.005

The following source data and figure supplement are available for figure 2:

**Source data 1.** Source data and summary statistics used for generating plots for *Figure 2*.
DOI: https://doi.org/10.7554/eLife.42819.007
**Source data 2.** Statistical results for *Figure 2* and *Figure 2—figure supplement 1*. numDF, numerator degrees of freedom. denDF, denominator degrees of freedom.
DOI: https://doi.org/10.7554/eLife.42819.008
**Figure supplement 1.** Baseline sleep bout analysis.
DOI: https://doi.org/10.7554/eLife.42819.006

interactions for bout number during REM sleep ($F_{(3,51)}$ = 2.832, p=0.047) and bout duration during NREM ($F_{(1.749,29.734)}$ = 4.077, p=0.032) and REM ($F_{(1.897,32.243)}$ = 6.536, p=0.005) sleep. Post-hoc comparisons showed these effects were driven by differences in the last half of the dark period (hours 19–24) when Shank3^ΔC mice had fewer REM bouts (p=0.003; *Figure 2—figure supplement 1A*) and shorter NREM (p=0.005) and REM (p=0.013) bouts (*Figure 2—figure supplement 1B*) than WT mice. Overall, these data show that Shank3^ΔC mice spend more time awake at the end of the dark phase compared to WT mice under baseline conditions.

Fourier analysis of the EEG indicates that Shank3$^{\Delta C}$ mice also sleep differently than WT mice (*Figure 2B*). Significant genotypic differences in spectral frequencies occur at points of non-overlapping 95% confidence intervals. During NREM sleep, power in the delta frequencies (0.5–4 Hz) was blunted and alpha frequencies (10–15 Hz) were enhanced for Shank3$^{\Delta C}$ mice relative to WT during the light and dark periods (*Figure 2B*). NREM delta power is a measure of synchrony of the neural network and can be a measure of sleep intensity or depth (*Achermann and Borbely, 2017*). These data indicate that Shank3$^{\Delta C}$ mice exhibit disrupted neural connectivity during NREM sleep which may suggest that Shank3$^{\Delta C}$ mice sleep less deeply under baseline conditions.

## Shank3$^{\Delta C}$ mice have abnormal homeostatic responses to sleep deprivation

To investigate sleep homeostasis in Shank3$^{\Delta C}$ mice we sleep deprived mutant and WT mice for 5 hr via gentle handling starting at light onset (hour 1). Sleep deprivation (SD) was effective resulting in WT and Shank3$^{\Delta C}$ mice spending 97.37 ± 0.76% and 95.08 ± 1.08% time in wakefulness, respectively, during the 5 hr SD period (t(18) = 1.732, p=0.100). Both WT and Shank3$^{\Delta C}$ mice showed homeostatic responses to SD (*Figure 3—figure supplement 1*) with reduced wakefulness and increased time spent in NREM and REM sleep during the recovery phase (*Figure 3—source data 2*). Sleep was also more consolidated after SD for both WT and Shank3$^{\Delta C}$ mice during the light period, and changes in sleep architecture did not statistically differ between WT and Shank3$^{\Delta C}$ mice (*Figure 3—figure supplement 2*). However, there were time x genotype interactions for wakefulness (F(18,288) = 2.025, p=0.009), NREM sleep (F(18,288) = 1.928, p=0.014), and REM sleep (F(18,288) = 1.716, p=0.036) over the SD recovery phase (hours 6–24). This effect was most pronounced near the end of the dark period where Shank3$^{\Delta C}$ mice spent more time in wakefulness and less time in NREM and REM sleep compared to WT (*Figure 3—figure supplement 1C*) which is consistent with the trend seen in the baseline data (*Figure 2A*). Changes in NREM EEG delta power were similar between WT and Shank3$^{\Delta C}$ mice, indicating that mutant mice accumulate sleep pressure similarly to WT mice (*Figure 3A*). However, Shank3$^{\Delta C}$ mice showed a transient enhancement of NREM EEG spectral power (F(1,18) = 12.07, p=0.003) compared to WT mice in the first 2 hr post-SD (hours 6–7). This enhancement was found in the higher frequencies (3.9–19.5 Hz; *Figure 3B*) and resolved to baseline and WT values during hours 11–12 (F(1,18) = 0.59, p=0.454; *Figure 3C*). These data suggest that some aspects of the Shank3$^{\Delta C}$ homeostatic response to sleep deprivation are abnormal.

Remarkably, Shank3$^{\Delta C}$ mice took longer to enter NREM sleep post-SD compared to WT (t (10.44) = −2.31, p=0.043; *Figure 3D*). As a consequence, Shank3$^{\Delta C}$ mice spent less time in NREM sleep compared to WT during the first 2 hr of the recovery phase (t(13.26) = 2.85, p=0.014; *Figure 3E*). During the subsequent 5 hr of the light period (hours 8–12), Shank3$^{\Delta C}$ mice spent more time in NREM sleep compared to WT (t(18) = −2.46; p=0.024; *Figure 3F*). Overall, our data show that Shank3$^{\Delta C}$ mice have difficulties falling asleep despite heightened sleep pressure.

## Shank3$^{\Delta C}$ mice show downregulation of circadian transcription factors in response to sleep deprivation

We conducted a genome-wide gene expression study to investigate the molecular basis for the Shank3$^{\Delta C}$ mouse sleep phenotype. Shank3$^{\Delta C}$ and WT adult male mice were subjected to 5 hr of SD starting at light onset and sacrificed immediately post-SD. Additional mice from both genotypes were sacrificed at the same time of day to determine differences in gene expression under home-cage conditions (HC). Prefrontal cortex was collected for all animals and subjected to RNA sequencing (RNA-seq, n = 5 per group). As expected, sleep deprivation is the greatest source of variation in the data (principal component 1), followed by the genotype effect (principal component 2; *Figure 4A*). Furthermore, the difference between genotypes is enhanced after sleep deprivation (*Figure 4A*) greatly increasing the number of differentially expressed genes between Shank3$^{\Delta C}$ and WT mice – starting at 69 genes (HC) and doubling to 134 genes (SD) (*Figure 4B*, *Figure 4—source data 1*; false discovery rate (FDR) < 0.1). Most of the differences in gene expression following SD are not present in HC conditions.

Clustering of gene expression patterns for all genes differentially expressed between Shank3$^{\Delta C}$ and WT mice reveals 3 groups of genes (*Figure 4C*). Cluster 1 contains genes that are downregulated in mutants versus WT mice under HC conditions. Genes in this cluster are also downregulated

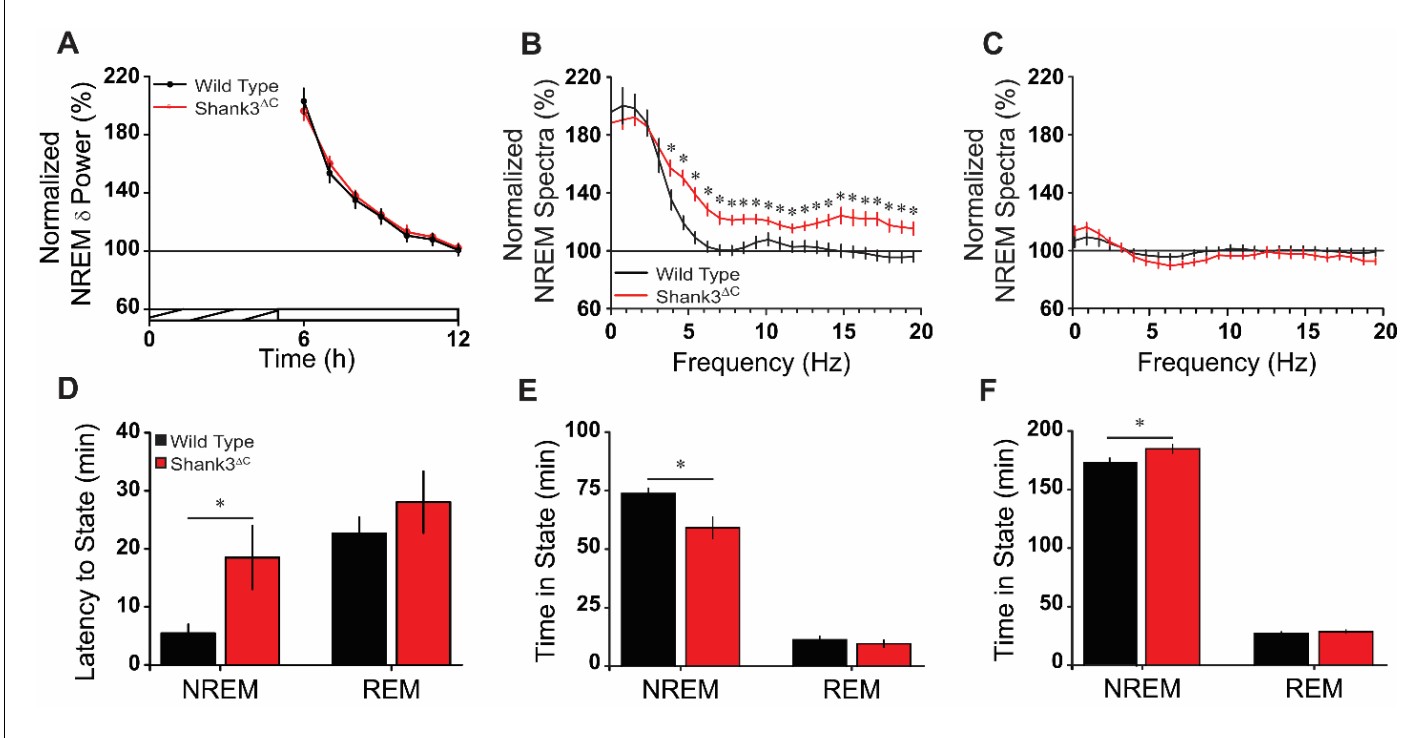

**Figure 3.** Shank3$^{\Delta C}$ mice take longer to fall asleep and sleep less after sleep deprivation. (**A**) Normalized NREM delta ($\delta$; 0.5–4 Hz) power during recovery sleep post-sleep deprivation. The cross-hatched bar on the x-axis denotes the 5 hr sleep deprivation period and the open bar denotes the remaining light period. Repeated measures ANOVA. (**B**) Normalized NREM spectra for the first 2 hr post-sleep deprivation (hours 6–7; significant from 3.9 to 19.5 Hz). One-way ANOVA with posthoc unpaired t-tests using Benjamini-Hochberg correction. (**C**) Normalized NREM spectra for the last 2 hr of the light period post-sleep deprivation (hours 11–12). One-way ANOVA. (**D**) Latency to enter NREM sleep and REM sleep after sleep deprivation. Unpaired t-test. (**E**) Time in NREM sleep and REM sleep for the first 2 hr post-sleep deprivation (hours 6–7). Unpaired t-test. (**F**) Time in NREM sleep and REM sleep for the remaining 5 hr of light period post-sleep deprivation (hours 8–12). Unpaired t-test. Values are means ± SEM for wild type (n = 10; black) and Shank3$^{\Delta C}$ (n = 10; red) mice. *p<0.05, difference from wild type.

DOI: https://doi.org/10.7554/eLife.42819.010

The following source data and figure supplements are available for figure 3:

**Source data 1.** Source data and summary statistics used for generating plots for *Figure 3*.
DOI: https://doi.org/10.7554/eLife.42819.013
**Source data 2.** Statistical results for *Figure 3*, *Figure 3—figure supplement 1*, and *Figure 3—figure supplement 2*.
DOI: https://doi.org/10.7554/eLife.42819.014
**Figure supplement 1.** Shank3$^{\Delta C}$ mice sleep less than wild type type after sleep deprivation.
DOI: https://doi.org/10.7554/eLife.42819.011
**Figure supplement 2.** Bout analysis after sleep deprivation.
DOI: https://doi.org/10.7554/eLife.42819.012

in response to SD in Shank3$^{\Delta C}$ mice but are generally unaffected by SD in WT mice. SD seems to exacerbate the difference between genotypes, which explains why differential expression of some of the genes in cluster 1 only reach statistical significance after SD. This cluster contains the majority of the genes that are differentially expressed between genotypes. Cluster 2 contains genes that are upregulated in mutants versus WT mice under HC conditions that are also downregulated by SD in WT mice. Cluster 3 contains genes that are normally upregulated by SD. For both clusters 2 and 3, the Shank3$^{\Delta C}$ mutation seems to dampen the response to SD.

To better understand the impact of the gene expression at the pathway level, we carried out functional annotation of the transcripts differentially expressed between Shank3$^{\Delta C}$ and WT mice (*Table 2*, *Table 2—source data 1*). Our results reveal that 1) MAPK/GnRH signaling and circadian rhythm-associated transcripts are downregulated in Shank3$^{\Delta C}$ mice and 2) sleep deprivation exacerbates that difference. Circadian transcription factors are particularly affected. For example, while

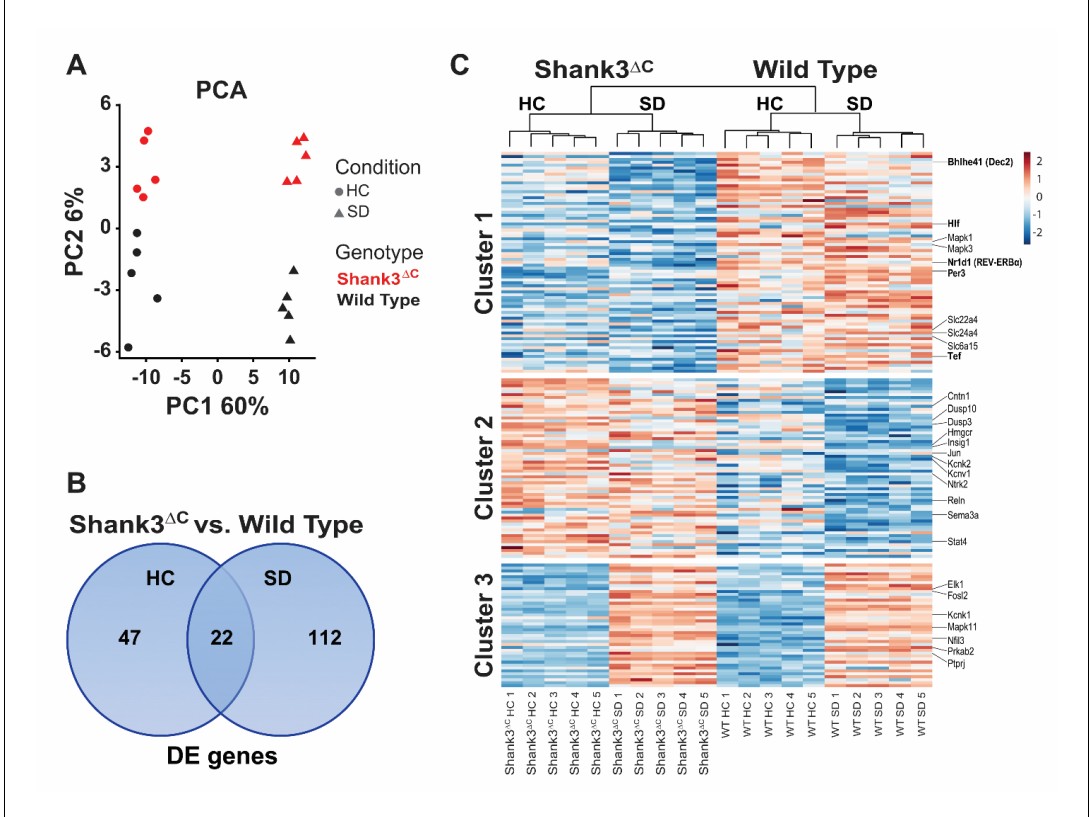

**Figure 4.** Sleep deprivation induces a two-fold difference in gene expression between Shank3$^{\Delta C}$ and wild type mice. RNA-seq study of gene expression from prefrontal cortex obtained from adult male Shank3$^{\Delta C}$ and wild type mice either under control homecage conditions (HC) or following 5 hr of sleep deprivation (SD). N = 5 mice per group. (**A**) Principal component analysis of normalized RNA-seq data shows that sleep deprivation is the main source of variance in the data (first principal component, PC1) and genotype is the second (second principal component, PC2). Percent variance explained by each PC is shown on each axis. (**B**) Venn diagram showing the number of genes differentially expressed at FDR < 0.1 between Shank3$^{\Delta C}$ and wild type mice in either control HC conditions or after SD. (**C**) Heat map of average scaled gene expression for all genes in (**B**). K-means clustering defined three clusters based on differences in gene expression across all comparisons. Genes belonging to the MAPK pathway and involved in circadian rhythms (see **Table 2**) are highlighted on the right.

DOI: https://doi.org/10.7554/eLife.42819.015

The following source data and figure supplement are available for figure 4:

**Source data 1.** Genes differentially expressed between Shank3$^{\Delta C}$ vs. wild type mice.
DOI: https://doi.org/10.7554/eLife.42819.017

**Figure supplement 1.** Recovery of positive control genes regulated by sleep deprivation.
DOI: https://doi.org/10.7554/eLife.42819.016

expression of *Per3*, *Hlf*, and *Tef* already differs under homecage conditions, SD leads to additional genotype-specific differences in expression of *Nr1d1* and *Bhlhe41*. All of the above mentioned circadian transcription factors belong to cluster 1 on our heat map (**Figure 4C**) showing downregulation in response to SD only in Shank3$^{\Delta C}$ mice.

## Wheel-running activity in Shank3$^{\Delta C}$ mice is impaired in constant darkness

The results of our RNA-seq analysis revealed a general downregulation of circadian transcription factors in the mutants. These findings suggested that Shank3$^{\Delta C}$ mice might also have abnormal circadian rhythms. To test this possibility, we measured wheel-running activity in Shank3$^{\Delta C}$ and WT mice. Wheel-running data were collected for 2 weeks under LD (LD weeks 2–3), then mice were released into constant darkness (DD) for 3 weeks (DD weeks 1–3; **Figure 5** and **Figure 5—figure supplement 1**). Alpha, the length of the active phase, was calculated as the time between activity onset and offset. Period length was the time elapsed from the start of an active phase to the start of the

**Table 2.** Results of functional bioinformatics analysis of the genes differentially expressed between Shank3$^{\Delta C}$ mice relative to wild type mice.

Functional annotation and clustering analysis was performed using DAVID (https://david.ncifcrf.gov) and functional information was obtained from the following databases: GO (Biological process and Molecular function), KEGG pathways, and Uniprot keywords. Enrichment was performed relative to all transcripts expressed in the mouse prefrontal cortex as defined by our RNA-seq data. Enriched functional terms were clustered at low stringency, to obtain clusters with enrichment score >1.2 (corresponding to an average p-value>0.05). See *Table 2—source data 1* for details.

**Shank3$^{\Delta C}$ vs. Wild Type**

| *Homecage* | *Sleep Deprivation* |
|---|---|
| Up-regulated | Up-regulated |
| Cholesterol Metabolism: *Hmgcr, Insig1* | Potassium Ion Transport: *Kcnv1, Kcnk1, Kcnk2* |
| Transcription: *Jun, Fosl2, Nfil3, Stat4* | Dephosphorylation: *Dusp10, Dusp3, Ptprj* |
|  | Neuron Projection Development: *Cntn1, Ntrk2, Reln, Sema3a* |
| Down-regulated | Down-regulated |
| MAPK Signaling: *Mapk3* (ERK1), *Elk1* | GnRH Signaling: *Mapk1* (ERK2), *Elk1, Mapk11* (p38) |
| Circadian Rhythms: *Per3, Tef, Hlf* | Circadian Rhythms: *Per3, Nr1d1* (REV-ERBα), *Tef, Hlf, Prkab2* (AMPK), *Bhlhe41* (DEC2) |
| Transcription: *Tef, Hlf* | Sodium Ion Transport: *Slc6a15, Slc22a4, Slc24a4* |

DOI: https://doi.org/10.7554/eLife.42819.018

The following source data is available for Table 2:

Source data 1. Functional annotation clustering analysis of genes in *Figure 4—Source data 1*.
DOI: https://doi.org/10.7554/eLife.42819.019

subsequent active phase. There was no differences between Shank3$^{\Delta C}$ and WT mice for alpha or period length during the DD period (*Table 3*), but Shank3$^{\Delta C}$ mice ran fewer revolutions than WT mice from LD week 3 – DD week 3 (F(1,338) = 30.96; p<0.0001; *Table 3*, *Table 3—source data 1*). Some Shank3$^{\Delta C}$ mice greatly reduced their running at certain periods of the DD period, which is reflected in the variance for period, alpha, and activity measures (*Figure 5*, *Figure 5—figure supplement 1*, and *Table 3*). Over 5 weeks of continuous LD conditions, however, wheel-running activity for Shank3$^{\Delta C}$ mice instead increased (F(1,542) = 85.99, p<0.0001; *Supplementary file 1*, *Source data 1*). Overall, these data indicate that constant darkness impairs wheel-running activity of Shank3$^{\Delta C}$ mice.

## Discussion

The present study is the first to establish a role for *Shank3* in mammalian sleep. We show that both PMS patients and *Shank3* mutant mice have trouble falling asleep (*Figures 1*, *2* and *3*). This phenotype is widely observed in the ASD population and until now had not been replicated in animal models. Shank3$^{\Delta C}$ mice have problems falling asleep after periods of extended wakefulness when sleep pressure is high, such as at the end of the baseline dark period (*Figure 2A*) or following sleep deprivation (*Figure 3D-E*). They also have blunted NREM delta power (*Figure 2B*). However, Shank3$^{\Delta C}$ mice accumulate sleep pressure (*Figure 3A*) and show no gross abnormalities in circadian rhythms (*Figure 5*). This suggests that the primary deficit is in sleep onset.

Our molecular studies show that sleep deprivation increases differences in gene expression between Shank3$^{\Delta C}$ and WT mice (*Figure 4*). These differences point to the downregulation of circadian transcription factors and genes involved in the MAPK/GnRH pathways in the mutants (*Table 2*). Circadian transcription factors affected include *Per3*, *Hlf*, *Tef*, *Nr1d1* (REV-ERBα), and *Bhlhe41* (DEC2). We therefore investigated the effects of our *Shank3* mutation in circadian rhythms. Shank3$^{\Delta C}$ mice do not show a disruption in circadian rhythmicity, but they do exhibit a large reduction in wheel-running activity in response to constant darkness (*Figure 5*, *Table 3*). Shank3$^{\Delta C}$ mice were reported to have deficits in motor coordination (*Kouser et al., 2013*); however, under a 12:12 hr LD schedule their wheel-running activity increases over time (*Supplementary file 1*, *Source data 1*). Daily rhythms of activity in rodents are linked to circadian oscillations in dopamine release (*Feenstra et al., 2000*; *Menon et al., 2019*) in the frontal cortex as well as in the striatum, a motor

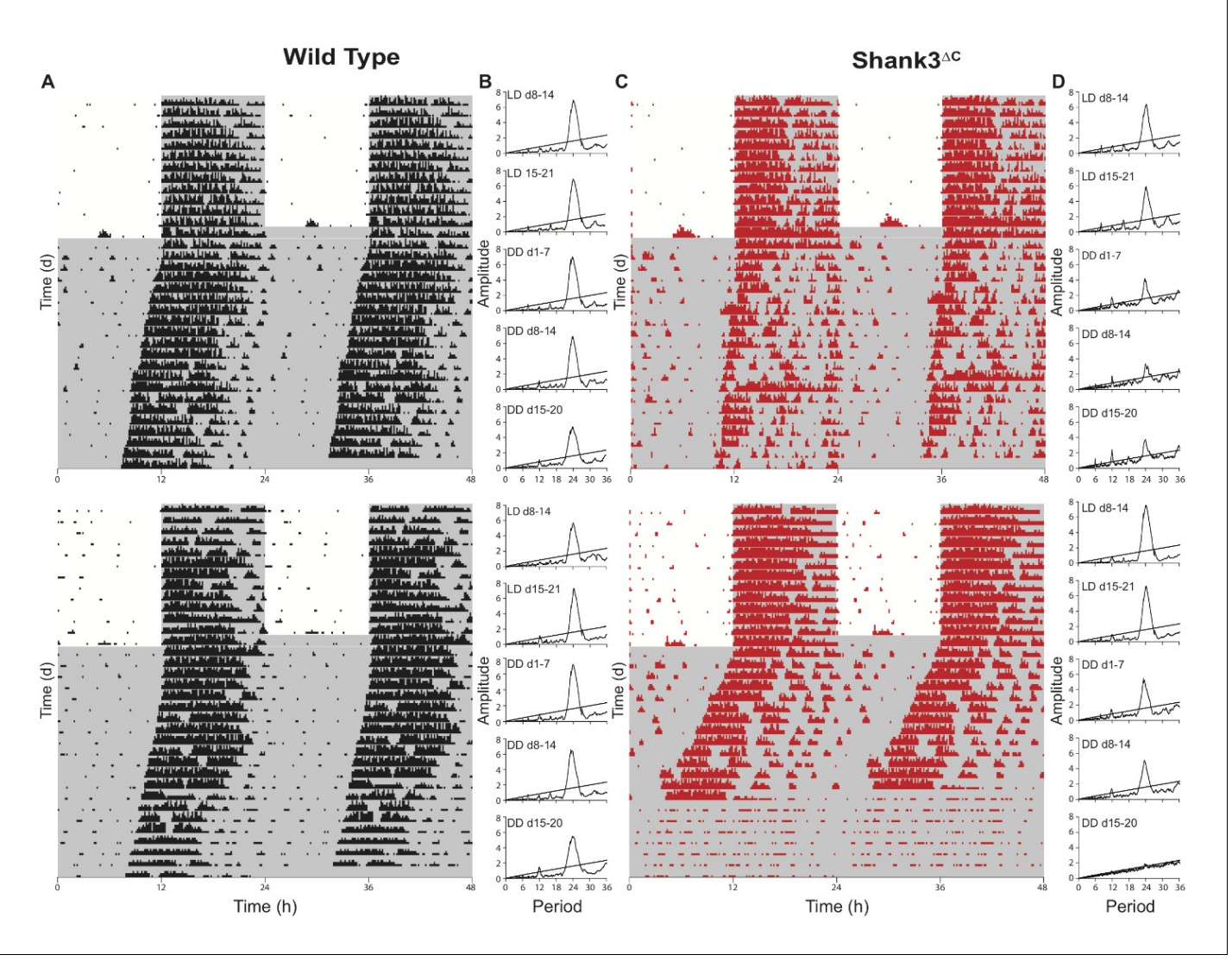

**Figure 5.** Shank3$^{\Delta C}$ mice show disruption of running wheel activity in constant darkness. Representative actograms and periodograms for two wild type and two Shank3$^{\Delta C}$ mice. Mice were entrained to a 12:12 hr light:dark cycle (LD, 559 ± 4 : 0 ± 0 lux) for two weeks prior to 3 weeks constant darkness (DD, 0 ± 0 lux). Gray shading is representative of the dark period. (**A**) Actograms for two wild type mice. (**B**) Corresponding periodograms for wild type mice. (**C**) Actograms for two Shank3$^{\Delta C}$ mice. (**D**) Corresponding periodograms for Shank3$^{\Delta C}$ mice.

DOI: https://doi.org/10.7554/eLife.42819.022

The following figure supplement is available for figure 5:

**Figure supplement 1.** Actograms for all wild type and Shank3$^{\Delta C}$ mice.

DOI: https://doi.org/10.7554/eLife.42819.023

region with high levels of *Shank3* expression (*Peça et al., 2011*). Together with a reduction in circadian gene expression, this DD-specific activity deficit suggests that the mutant sleep phenotype involves clock gene functions outside of their central time-keeping role. Interestingly, mutations in some of the circadian transcription factors we identified, *Per3* and *Bhlhe41* (DEC2), indeed lead to deficits in sleep regulation (*Archer et al., 2018*; *He et al., 2009*; *Hirano et al., 2018*). Our results support a role of clock genes in influencing sleep outside of their roles in generating circadian rhythms (*Franken, 2013*).

An interesting question is how can deletion of exon 21 of *Shank3* lead to dysregulation of transcriptional and signaling pathways linked to sleep and sleep loss? Exon 21 of *Shank3* encodes the homer and cortactin interaction domains of the protein. Homer interacts with metabotropic

**Table 3.** Shank3$^{\Delta C}$ mice show reduced wheel-running activity during constant darkness.
Summary of wheel-running behavior measured for each week of 12:12 hr light dark cycle (LD, 559 ± 4 : 0 ± 0 lux) or constant darkness (DD, 0 ± 0 lux) for both wild type and Shank3$^{\Delta C}$ mice. A linear mixed-effects model was used to estimate the contributions of genotype and the interaction between genotype and time for period, alpha, and wheel running activity for LD week three through DD week 3. There is a significant interaction effect for genotype x time for activity (p < 0.0001; indicated by *) but no significant effect for genotype x time for period (p = 0.15) or genotype x time for alpha (p = 0.88). See *Table 3—source data 1* for details. Values are means ± SEM for wild type (n = 8) and Shank3$^{\Delta C}$ (n = 7) mice. Significance at p < 0.05.

| | *Wild Type* | *Shank3$^{\Delta C}$* |
|---|---|---|
| | *Period (h)* | |
| LD Week 2 | 24.0 ± 0.0 | 24.0 ± 0.0 |
| LD Week 3 | 24.0 ± 0.0 | 23.9 ± 0.1 |
| DD Week 1 | 23.9 ± 0.1 | 23.7 ± 0.1 |
| DD Week 2 | 23.7 ± 0.1 | 20.3 ± 3.4 |
| DD Week 3 | 23.5 ± 0.2 | 25.8 ± 1.5 |
| | *Alpha (h)* | |
| LD Week 2 | 10.0 ± 0.4 | 10.7 ± 0.4 |
| LD Week 3 | 10.3 ± 0.4 | 10.7 ± 0.5 |
| DD Week 1 | 10.5 ± 0.2 | 9.0 ± 0.4 |
| DD Week 2 | 10.4 ± 0.4 | 8.2 ± 0.7 |
| DD Week 3 | 9.0 ± 1.0 | 9.9 ± 1.0 |
| | *Activity (rev/day)** | |
| LD Week 2 | 33985.2 ± 565.9 | 23258.2 ± 861.2 |
| LD Week 3 | 40306.7 ± 1257.1 | 25365.1 ± 964.0 |
| DD Week 1 | 37942.6 ± 795.4 | 23027.7 ± 669.6 |
| DD Week 2 | 38250.7 ± 431.1 | 16820.7 ± 1302.5 |
| DD Week 3 | 38586.3 ± 231.1 | 6961.5 ± 383.9 |

DOI: https://doi.org/10.7554/eLife.42819.020
The following source data is available for Table 3:
**Source data 1.** Results of genotype x time interaction analysis for LD week three through DD week 3.
Linear mixed-effects ANOVA for LD week three through DD week 3. The model was fitted using genotype, time, and their interaction as fixed effects and the individual mice labeled as random effect. There is a significant effect of genotype x time for activity (p<0.0001) and no significant effect for genotype x time or genotype for period and alpha. Significance at p<0.05. numDF, numerator degrees of freedom. denDF, denominator degrees of freedom.
DOI: https://doi.org/10.7554/eLife.42819.021

glutamate receptors (mGluRs) and SHANK3/homer complexes anchor mGluRs to the synapse. Shank3$^{\Delta C}$ mice show a marked reduction of the major isoforms of SHANK3 as well as an increase of mGluRs at the synapse (*Kouser et al., 2013*). mGluR signaling activates the MAPK/ERK pathway, a key regulator of activity-dependent transcription and synaptic plasticity in mature neurons (*Thomas and Huganir, 2004*). So the role of SHANK3 at the synaptic membrane explains the observed regulation of MAPK pathway genes (*Table 2*). However, it is not yet clear how SHANK3 regulates expression of circadian transcription factors within the nucleus. One possibility might include the role of SHANK3 in Wnt signaling. SHANK3 modulates Wnt-mediated transcriptional regulation by regulating internalization of Wnt receptor Frizzled-2 (*Harris et al., 2016*) and nuclear translocation of the Wnt ligand beta-catenin (*Qin et al., 2018*). The Wnt pathway kinase GSK3β phosphorylates circadian transcription factors PER2 (*Iitaka et al., 2005*), CRY2 (*Harada et al., 2005*), and REV-ERBα (*Yin et al., 2006*); however, this mechanism modulates circadian period length which is not altered in Shank3$^{\Delta C}$ mice. A second potential mechanism is nuclear translocation of SHANK3 itself. SHANK3 is known to undergo synaptic-nuclear shuttling in response to neuronal

activity and interact with nuclear ribonucleoproteins and components of the RNA Pol II mediator complex (*Grabrucker et al., 2014*). Deletion of the C-terminus leads to nuclear accumulation of SHANK3 and alterations in gene expression (*Cochoy et al., 2015*; *Grabrucker et al., 2014*). Thus, mutations in exon 21 of *Shank3* could lead to deficits in transcriptional regulation in response to sleep deprivation through direct regulation of transcription in the nucleus. Yeast two-hybrid data show that SHANK3 can directly bind the circadian transcription factors REV-ERBα (encoded by *Nr1d1*) and DEC1 (encoded by *Bhlhe40*), a close paralog of DEC2 (*Bhlhe41*) (*Sakai et al., 2011*). Future studies of the effects of sleep on SHANK3 nuclear translocation will provide new insights into the non-synaptic function of shank proteins and their role in sleep and circadian rhythms.

# Materials and methods

## Key resources table

| Reagent type (species) or resource | Designation | Source or reference | Identifiers |
| --- | --- | --- | --- |
| Gene (*Mus musculus*) | *Shank3; SH3 and multiple ankyrin repeat domains 3* | NA | ENTREZ_ID: 58234 |
| Gene (*Mus musculus*) | *Per3; period circadian clock 3* | NA | ENTREZ_ID: 18628 |
| Gene (*Mus musculus*) | *Hlf; hepatic leukemia factor* | NA | ENTREZ_ID: 217082 |
| Gene (*Mus musculus*) | *Tef; thyrotroph embryonic factor* | NA | ENTREZ_ID: 21685 |
| Gene (*Mus musculus*) | *Nr1d1; nuclear receptor subfamily 1, group D, member 1;* | NA | ENTREZ_ID: 217166 |
| Gene (*Mus musculus*) | *Bhlhe41; basic helix-loop-helix family, member e41;* | NA | ENTREZ_ID: 79362 |
| Strain, strain background (*Mus musculus*) | Shank3$^{\Delta C}$; Shank3tm1.1Pfw/J | *Kouser et al., 2013*; The Jackson Laboratory | RRID:IMSR_JAX:018398 |
| Commercial assay or kit | RNAeasy kit | Qiagen | 74104 |
| Commercial assay or kit | High Sensitivity RNA Analysis Kit | Advanced Analytical Technologies | DNF-472 |
| Commercial assay or kit | High Sensitivity NGS Fragment Analysis Kit | Advanced Analytical Technologies | DNF-474 |
| Commercial assay or kit | KAPA Library Quantification Kit | Kapabiosystems | KK4824 |
| Commercial assay or kit | TruSeq Stranded mRNA Library Prep Kit | Illumina | RS-122–2101 |
| Software, algorithm | HiSeq Control Software version (v. 2.2.68) | Illumina | 15073358 Rev A |

*Continued on next page*

*Continued*

| Reagent type (species) or resource | Designation | Source or reference | Identifiers |
|---|---|---|---|
| Software, algorithm | bcl2fastq (v. 2.17.1.14) | R/Bioconductor | SCR_015058 |
| Software, algorithm | Rsubread (v. 1.26.1) | R/Bioconductor | SCR_016945 |
| Software, algorithm | NOISeq (v.2.22.1) | R/Bioconductor | SCR_003002 |
| Software, algorithm | RUVSeq (v. 1.12.0) | R/Bioconductor | SCR_006263 |
| Software, algorithm | edgeR (v. 3.20.9) | R/Bioconductor | SCR_012802 |
| Software, algorithm | R package nlme (v. 3.5.0) | CRAN | SCR_015655 |
| Software, algorithm | R package mgcf (v. 3.5.0) | CRAN | |
| Software, algorithm | R package pheatmap (v. 1.0.10) | CRAN | SCR_016418 |
| Software, algorithm | Database for annotation, visualization, and integrated discovery (v 6.7) | DAVID; https://david.ncifcrf.gov | |
| Software, algorithm | VitalRecorder | Kissei Comtec | 3, 0, 0, 0 |
| Software, algorithm | SleepSign for Animal | Kissei Comtec | 3, 0, 0, 812 |
| Software, algorithm | Wheel manager | Med Associates Inc | SOF-860 |
| Software, algorithm | ClockLab (v. 3.5.0) | Actimetrics | 6.0.50 |

## Sleep questionnaire study

Sleep questionnaire data for PMS patients was obtained through a data access agreement with the PMSIR. The PMSIR contains demographic, clinical, and genetic data on PMS patients. Parents and caregivers are able to create a profile on behalf of the patient, enter demographic data, complete questionnaires on symptoms and development, and upload files such as genetic test reports and other medical records. Trained genetic counselors curate all genetic data to ensure each patient has as complete and accurate 'genetic profile' insofar as possible. Researchers may apply for data exports containing de-identified clinical data, developmental data, and genetic data. Once approved, the appropriate search is performed in the database and the results are provided to the researcher. Data in this study corresponds to a database export performed on 12/1/2016 containing the results of a sleep questionnaire completed by caregivers, as well as biographic and genetic information of PMS individuals. Difficulty falling asleep is defined as needing more than an hour to fall asleep. Multiple night awakenings is defined as more than two awakenings. Reduced sleep time is defined as less than 6 hr per night. Parasomnias are defined as abnormal movements, behaviors, emotions, perceptions, and dreams that occur while falling asleep or sleeping. Presence of sleep apnea is defined as having received a diagnosis of sleep apnea. Only individuals with a genetic counselor-confirmed deletion of the *SHANK3* gene were included in the analysis. The final dataset included 176 individuals – 78 males and 98 females (age range 1–39 years).

## Animals

Heterozygous Shank3$^{+/\Delta C}$ mice were obtained from Dr. Paul Worley at Johns Hopkins University. Shank3$^{+/\Delta C}$ breeding pairs were established to obtain wild type (WT) and Shank3$^{\Delta C}$ littermates. Mice were housed in standard cages at 24 ± 1°C on a 12:12 hr light:dark cycle (unless otherwise specified) with food and water *ad libitum*. All experimental procedures were approved by the Institutional Animal Care and Use Committee of Washington State University and conducted in accordance with National Research Council guidelines and regulations for experiments in live animals.

## Assessment of sleep-wake behavior

### Surgical procedures

Adult male mice (12-weeks-old) were stereotaxically implanted with electroencephalographic (EEG) and electromyographic (EMG) electrodes under isoflurane anesthesia according to previously published methods (*Frank et al., 2002*). Briefly, four stainless steel screws (BC-002MPU188, Bellcan International Corp, Hialeah, FL) were placed contralaterally over frontal (2) and parietal (2) cortices, and 2 EMG electrodes were inserted in the nuchal muscles. Mice were allowed 5 days of recovery from surgery prior to habituation to the recording environment.

### Experimental design

Six days after surgery, wild type (WT; n = 10) and Shank3$^{\Delta C}$ (n = 10) littermates were connected to a lightweight, flexible tether and allowed 5 days to habituate to the tether and recording environment. After habituation, mice underwent 24 hr undisturbed baseline EEG and EMG recording starting at light onset (hour 1). The next day, mice were sleep deprived for 5 hr (hours 1–5) via gentle handling beginning at light onset according to previously published methods (*Halassa et al., 2009*; *Vecsey et al., 2012*; *Ingiosi et al., 2015*). Mice were then allowed 19 hr undisturbed recovery sleep (hours 6–12 of the light period and hours 13–24 of the dark period).

### EEG/EMG data acquisition and analysis

EEG and EMG data were collected with Grass 7 polygraph hardware (Natus Medical Incorporated, Pleasanton, CA) via a light-weight, counterbalanced cable, amplified, and digitized at 200 Hz using VitalRecorder acquisition software (SleepSign for Animal, Kissei Comtec Co., LTD, Nagano, Japan). EEG and EMG data were high- and low-pass filtered at 0.3 and 100 Hz and 10 and 100 Hz, respectively.

Wakefulness and sleep states were determined by visual inspection of the EEG waveform, EMG activity, and fast Fourier transform (FFT) analysis by an experimenter blinded to the genotype. Data were scored as wakefulness, NREM sleep, or REM sleep with 4 s resolution using SleepSign for Animal as previously described (*Halassa et al., 2009*).

The EEG was subjected to fast Fourier transform analysis to produce power spectra between 0–20 Hz with 0.781 Hz resolution. Delta (δ) was defined as 0.5–4 Hz, theta (θ) as 5–9 Hz, and alpha (α) as 10–15 Hz. For genotypic comparisons of light and dark period spectral data, each spectral bin was expressed as a percentage of the sum of total state-specific EEG power (0–20 Hz) of the baseline 12 hr light period and 12 hr dark period, respectively (*Baracchi and Opp, 2008*). The relation between EEG spectral power and frequency under baseline conditions was fit to smooth curves and 95% confidence intervals for the respective light and dark periods. The analysis was performed via Generalized Additive Models (GAM) (*Hastie and Tibshirani, 1990*) using the R package mgcf (v. 3.5.0). For hourly NREM delta power analysis after sleep deprivation, spectral values within the delta band for each hour were normalized to the average NREM delta band value of the last 4 hr of the baseline light period (hours 9–12) and expressed as a percentage (*Franken et al., 2001*). Changes in individual NREM EEG spectral bins (0–20 Hz) after sleep deprivation were normalized to corresponding baseline spectral bins obtained from mean values from the last 4 hr of the baseline light period (hours 9–12). EEG epochs containing artifacts were excluded from spectral analyses.

Statistical analyses were performed using SPSS for Windows (IBM Corporation, Armonk, NY) and R statistical language. Data are presented as means ± standard error of the mean (SEM). If an individual hour bin did not contain a full hour of data due to technical issues (e.g. removal of recording tether), that hour was excluded from analysis for the corresponding individual animal. A general linear model for repeated measures (RM) using time (hours or period) as the repeated measure and

genotype (WT vs. Shank3$^{\Delta C}$) or treatment (undisturbed baseline vs. sleep deprivation) as the between subjects factor was used when multiple measurements were made over time (i.e. total sleep time, time in stage, bout number, bout duration, hourly NREM delta power). Repeated measures were tested for sphericity, and a Greenhouse-Geisser correction was applied when appropriate. Posthoc pairwise comparisons using Sidak corrections were performed when there were significant interaction effects or main effects of genotype or treatment. Baseline time in state and bout data RM comparisons were made over all time intervals during the full 24 hr recording period (hours 1–24). For sleep deprivation experiments, time in state and bout data RM comparisons were made over all time intervals during the full recovery period (hours 6–24). Hourly NREM delta power RM comparisons post-sleep deprivation were made over hours 6–12 of the recovery phase. Comparisons of normalized NREM EEG spectral power changes after sleep deprivation were made using one-way ANOVA with total spectral power as the dependent variable and genotype as the between subjects factor. If the one-way ANOVA yielded a significant result, unpaired Student's t-tests with Benjamini-Hochberg correction were then used for comparisons of individual NREM EEG spectral frequency bins with genotype as the grouping variable. Unpaired Student's t-tests with genotype as the grouping variable were used to analyze latency to state and total time in state post-sleep deprivation. Latency to NREM or REM sleep after sleep deprivation was defined as time elapsed from release to recovery sleep to the first bout of NREM sleep (bout $\geq$28 s or seven consecutive epochs) or REM sleep (bout $\geq$16 s or four consecutive epochs).

## Genome-wide gene expression

### RNA isolation

Adult male (8–10 week-old) Shank3$^{\Delta C}$ mice and WT littermates were divided into two groups: homecage controls (WT n = 5; Shank3$^{\Delta C}$n = 5) and sleep deprived (WT n = 5; Shank3$^{\Delta C}$n = 5). All mice were individually housed. Homecage control mice were left undisturbed and sacrificed 5 hr after light onset (hour 6). Mice in the sleep deprived group were sleep deprived for 5 hr via gentle handling starting at light onset (hour 1) and then sacrificed upon completion of sleep deprivation (hour 6) without allowing for any recovery sleep. Mice were sacrificed by live cervical dislocation (alternating between homecage controls and sleep deprived mice), decapitated, and prefrontal cortex was swiftly dissected on a cold block. Tissue was flash frozen in liquid nitrogen and stored at −80°C until processing (*Poplawski et al., 2016*; *Tudor et al., 2016*). To avoid day-specific effects on gene expression, this protocol was repeated over a 5 day period, and one sample from each genotype + treatment group was collected per day. Tissue was collected within the first 15 min of the hour. All tissue samples were later homogenized in Qiazol buffer (Qiagen, Hilden, Germany) using a TissueLyser (Qiagen) and all RNA was extracted using the Qiagen RNAeasy kit (Qiagen) on the same day.

### RNA-seq library preparation and sequencing

The integrity of total RNA was assessed using Fragment Analyzer (Advanced Analytical Technologies, Inc, Ankeny, IA) with the High Sensitivity RNA Analysis Kit (Advanced Analytical Technologies, Inc). RNA Quality Numbers (RQNs) from 1 to 10 were assigned to each sample to indicate its integrity or quality. '10' stands for a perfect RNA sample without any degradation, whereas '1' marks a completely degraded sample. All RNA samples had RQNs ranging from 8.8 to 10 and were used for RNA library preparation with the TruSeq Stranded mRNA Library Prep Kit (Illumina, San Diego, CA). Briefly, mRNA was isolated from 2.5 µg of total RNA using poly-T oligo attached to magnetic beads and then subjected to fragmentation, followed by cDNA synthesis, dA-tailing, adaptor ligation, and PCR enrichment. The sizes of RNA libraries were assessed by Fragment Analyzer with the High Sensitivity NGS Fragment Analysis Kit (Advanced Analytical Technologies, Inc). The concentrations of RNA libraries were measured by StepOnePlus Real-Time PCR System (ThermoFisher Scientific, San Jose, CA) with the KAPA Library Quantification Kit (Kapabiosystems, Wilmington, MA). The libraries were diluted to 2 nM with Tris buffer (10 mM Tris-HCl, pH 8.5) and denatured with 0.1 N NaOH. Eighteen pM libraries were clustered in a high-output flow cell using HiSeq Cluster Kit v4 on a cBot (Illumina). After cluster generation, the flow cell was loaded onto HiSeq 2500 for sequencing using HiSeq SBS kit v4 (Illumina). DNA was sequenced from both ends (paired-end) with a read length of 100 bp.

## RNA-seq data analysis

HiSeq Control Software version 2.2.68 was used for base calling. The raw bcl files were converted to fastq files using the software program bcl2fastq2.17.1.14. Adaptors were trimmed from the fastq files during the conversion. The average sequencing depth for all samples was 52 million read pairs. Sequenced reads were mapped with GSNAP (parameter = -N 1 m 7 -A sam –nofails) to mm10. On average 84% of the sequenced reads mapped uniquely to the mm10 genome. All statistical analyses were performed using open source software freely available through the R/Bioconductor project (*Gentleman et al., 2004*). The count matrix was obtained using featureCounts (parameters = isPairedEnd = TRUE, requireBothEndsMapped = TRUE, annot.inbuilt = 'mm10') from the package Rsubread (v. 1.26.1) under Bioconductor (version 3.6) with R (v. 3.4.1) using mm10 built-in annotations (Mus_musculus.GRCm38.90) with ENTREZ ID as row names. Gene counts were filtered using a proportion test (counts per million cutoff of 1), as implemented in the NOISeq package (v.2.22.1) (*Tarazona et al., 2015*). Removal of unwanted variation (RUV) normalization was performed using RUVSeq (v. 1.12.0) (*Risso et al., 2014*; *Peixoto et al., 2015*) after the data was normalized by Trimmed Mean of M-values (TMM) (*Robinson and Oshlack, 2010*) using edgeR (v. 3.20.9) (*Robinson et al., 2010*). Specifically, RUVs (with k = 5) was used after defining groups based on both genotype and treatment and using a list of 2677 negative control genes obtained as genes with an adjusted p-value>0.9 in the comparison between sleep deprivation and controls in microarray data available through GEO (GSE78215) (obtained from *Gerstner et al., 2016*). Analogously, a list of 579 positive control genes obtained from the same study were used to evaluate the ability of our RNA-seq study to replicate known differences in gene expression. These positive controls were genes found to be differentially expressed (FDR < 0.005) by 5 hr of sleep deprivation relative to homecage controls in WT mice in four independent studies using two different microarray platforms and two different brain regions (Additional file 2 from *Gerstner et al., 2016*). *Figure 4—figure supplement 1* shows that our RNA-seq study is able to recover 78% of the positive controls as differentially expressed when performing the same comparison done in the previous studies (HC5 vs SD5 in WT), while being able to detect many more differences in gene expression. This is a very high level of concordance considering these controls were obtained using a different technology and data were collected from experiments in four independent laboratories. Therefore, we are confident in the ability of our study to detect differences in gene expression that can be independently validated.

Differential expression analysis was performed using edgeR (v. 3.20.9) with a factorial design that included genotype (WT or Shank3$^{\Delta C}$) and treatment (homecage control or sleep deprivation). We specified the following contrasts: wild type sleep deprived vs. wild type homecage controls (WTSD vs. WTHC), Shank3$^{\Delta C}$ sleep deprived vs. Shank3$^{\Delta C}$ homecage controls (S3SD vs. S3HC), Shank3$^{\Delta C}$ homecage controls vs. wild type homecage controls (S3HC vs. WTHC), and Shank3$^{\Delta C}$ sleep deprived vs. wild type sleep deprived (S3SD vs. WTSD). To gain insight on the different effects of sleep deprivation between wild type and mutant mice, we considered the union of the differentially expressed genes between genotypes both in homecage controls and sleep deprived mice. We clustered the genes using k-means (k = 3) and the samples using hierarchical clustering based on the selected genes. The resulting ordered genes and samples were visualized as a heatmap using the pheatmap package (v. 1.0.10). The R code used for RNA-seq data analysis is publicly available at https://github.com/drighelli/peixoto (*Righelli et al., 2019*; copy archived at https://github.com/elifesciences-publications/peixoto).

Functional annotation analysis of genes differentially expressed at FDR < 0.1 was based on Ensembl Gene IDs and performed using the database for annotation, visualization, and integrated discovery v 6.7 (DAVID, https://david.ncifcrf.gov). The following functional categories were used: GO Biological Process, GO Molecular Function, Uniprot keywords, and KEGG pathways. Enrichment was determined relative to all genes expressed in the mouse prefrontal cortex based on our data. We used an enrichment p-value (EASE) cutoff <0.1 for individual functional terms. Enriched functional terms were clustered at low stringency, to obtain clusters with an enrichment score >1.2 (corresponding to an average EASE >0.05).

## Assessment of circadian rhythms

### Experimental design

Adult male (9-week-old) wild type (n = 8) and Shank3$^{\Delta C}$ (n = 7) littermate mice were individually housed with a low-profile wireless running wheel (Med Associates Inc, Fairfax, VT) and kept under 12:12 hr light:dark (LD) cycle for 3 weeks (including 1 week of habituation) followed by 3 weeks of complete darkness (DD). Two Shank3$^{\Delta C}$ mice from a starting n of 9 were excluded from the experiments because they failed run on the wheel for the duration of the experiment. The experiment was repeated with an independent cohort of 11-week-old male wild type (n = 8) and Shank3$^{\Delta C}$ (n = 8) littermate mice kept under a 12:12 hr light:dark (LD) cycle for one week of habituation and then 5 weeks of data collection. One Shank3$^{\Delta C}$ mouse from a starting n of 9 was excluded from the second experiment because it failed to run on the wheel for the duration of the experiment. Daily checks were made to verify that each running wheel was transmitting data and that no light was detected in the room during constant darkness conditions.

### Data acquisition and analysis

Wheel-running behavior was continuously monitored using Wheel Manager Software and exported into 10 min time bins (Med Associates Inc). ClockLab version 6.0.50 (Actimetrics, Wilmette, IL) was used for analysis. Period, the circadian length in time from activity onset to the next activity onset, was calculated using ClockLab's chi-squared periodogram analysis function. Alpha, the length of the activity period, was calculated as the time from activity onset to offset (defined by manufacturer defaults). Activity was calculated as the total sum of wheel revolutions per day. A linear mixed-effects model ANOVA was used to estimate the contributions of genotype and the interaction between genotype and time for period, alpha, and wheel running activity for LD week three through DD week three in the first experiment, and LD week 1–5 in the second experiment. The model was fitted using genotype, time, and their interaction as fixed effects, and the individual mice were labeled as a random effect. The analysis was conducted with the R package nlme (v. 3.5.0) and is publicly available at https://github.com/drighelli/peixoto (*Righelli et al., 2019*; copy archived at https://github.com/elifesciences-publications/peixoto).

## Acknowledgements

Data used in the preparation of this article were obtained from the Phelan-McDermid Syndrome International Registry (PMSIR) version dated 12/1/2016.

We thank Dr. Paul Worley for providing the Shank3 mutant mouse line. We thank Dr. Hans Van Dongen and Dr. John Hogenesch for valuable discussion of the results. We thank Dr. Claudia Angelini for valuable discussion of the statistical methods.

## Additional information

### Funding

| Funder | Grant reference number | Author |
| --- | --- | --- |
| National Institute of Neurological Disorders and Stroke | K01NS104172 | Lucia Peixoto |

The funders had no role in study design, data collection and interpretation, or the decision to submit the work for publication.

### Author contributions

Ashley M Ingiosi, Conceptualization, Formal analysis, Investigation, Visualization, Methodology, Writing—original draft, Writing—review and editing; Hannah Schoch, Data curation, Formal analysis, Investigation, Visualization, Writing—review and editing; Taylor Wintler, Formal analysis, Investigation, Visualization, Methodology, Writing—original draft, Writing—review and editing; Kristan G Singletary, Investigation, Project administration, Writing—review and editing; Dario Righelli, Formal analysis, Visualization, Methodology, Writing—review and editing; Leandro G Roser, Data curation,

Formal analysis, Methodology, Writing—review and editing; Elizabeth Medina, Investigation, Methodology; Davide Risso, Conceptualization, Supervision, Visualization, Methodology, Writing—review and editing; Marcos G Frank, Conceptualization, Resources, Supervision, Methodology, Writing—review and editing; Lucia Peixoto, Conceptualization, Resources, Data curation, Formal analysis, Supervision, Funding acquisition, Investigation, Visualization, Methodology, Writing—original draft, Project administration, Writing—review and editing

#### Author ORCIDs
Ashley M Ingiosi (iD) https://orcid.org/0000-0003-3035-3010
Dario Righelli (iD) http://orcid.org/0000-0003-1504-3583
Davide Risso (iD) http://orcid.org/0000-0001-8508-5012
Lucia Peixoto (iD) https://orcid.org/0000-0002-8444-9600

#### Ethics
Human subjects: PMSIR patient data was obtained de-identified under IRB exemption 15005-Peixoto
Animal experimentation: This study was performed in strict accordance with the recommendations in the Guide for the Care and Use of Laboratory Animals of the National Institutes of Health. All of the animals were handled according to approved institutional animal care and use committee (IACUC) protocols of Washington State University. Protocol numbers: 04705-001 and 04704-001 IACUC 6155-Peixoto Breeding IACUC 4705-Peixoto experimental IACUC 4581-Frank experimental

#### Decision letter and Author response
Decision letter https://doi.org/10.7554/eLife.42819.033
Author response https://doi.org/10.7554/eLife.42819.034

## Additional files

#### Supplementary files
• Source data 1. Results of genotype x time interaction analysis for 5 weeks of LD. Linear mixed-effects ANOVA for LD week 1–5. The model was fitted using genotype, time, and their interaction as fixed effects and the individual mice labeled as random effect. There is a significant effect for genotype x time in activity (p<0.0001), a significant effect of genotype for period (p=0.047), and no significant effect for genotype x time or genotype for alpha. Significance at p<0.05. numDF, numerator degrees of freedom. denDF, denominator degrees of freedom.
DOI: https://doi.org/10.7554/eLife.42819.024

• Source code 1. Detailed report of statistical analysis including R code to reproduce the RNA-seq and circadian wheel running analysis.
DOI: https://doi.org/10.7554/eLife.42819.025

• Supplementary file 1. Wheel-running behavior in Shank3$^{\Delta C}$ mice across 5 weeks of continuous light: dark (LD) cycles. Summary of wheel-running behavior measured for each week of 12:12 hr light:dark cycles for both wild type and Shank3$^{\Delta C}$ mice. A linear mixed-effects ANOVA was used to estimate the contributions of genotype and the interaction between genotype and time for period, alpha, and wheel-running activity for LD week 1 through 5. There is no significant interaction effect for genotype x time for period (p=0.41), but there is a significant effect of genotype (p=0.047, indicated by *). There is no significant interaction effect for genotype x time for alpha (p=0.43), but there is a significant effect of time (p=0.004, indicated by [†]). There is a significant interaction effect (indicated by vertical bar) for genotype x time for activity (p<0.0001). See *Source data 1* for details. Values are means ± SEM for wild type (n = 8) and Shank3$^{\Delta C}$ (n = 8) mice.
DOI: https://doi.org/10.7554/eLife.42819.026

• Transparent reporting form
DOI: https://doi.org/10.7554/eLife.42819.027

## Data availability

Sequencing data have been deposited in GEO under accession code GSE113754. Source data files have been provided for Figures 1-4, Tables 1 and 2, and Supplementary Files 1 and 2. The R code used in this article is available on GitHub (https://github.com/drighelli/peixoto; copy archived at https://github.com/elifesciences-publications/peixoto). The R code used for the statistical analysis of RNA-seq and circadian wheel running data is also available in Source Code file 1.

The following dataset was generated:

| Author(s) | Year | Dataset title | Dataset URL | Database and Identifier |
|---|---|---|---|---|
| Peixoto L, Roser L, Schoch H, Wintler T | 2018 | RNA-Seq analysis of Sleep deprivation in wildtype and Shank3 mutant mice | https://www.ncbi.nlm.nih.gov/geo/query/acc.cgi?acc=GSE113754 | NCBI Gene Expression Omnibus, GSE113754 |

The following previously published dataset was used:

| Author(s) | Year | Dataset title | Dataset URL | Database and Identifier |
|---|---|---|---|---|
| Peixoto L, Frank M, Gerstner J | 2016 | Gene expression linked to sleep homeostasis in murine cortex | https://www.ncbi.nlm.nih.gov/geo/query/acc.cgi?acc=GSE78215 | NCBI Gene Expression Omnibus, GSE78215 |

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
