## [Decision Letter]

Thank you for submitting your article "Shank3 Modulates Sleep and Expression of Circadian Transcription Factors" for consideration by *eLife*. Your article has been reviewed by three peer reviewers, and the evaluation has been overseen by a Reviewing Editor and Huda Zoghbi as the Senior Editor. The following individual involved in the review of your submission has agreed to reveal her identity: Valerie Mongrain (Reviewer #3).

The reviewers have discussed the reviews with one another and the Reviewing Editor has drafted this decision to help you prepare a revised submission.

Summary:

The manuscript by Ingiosi et al., have reported the sleep problems in mice lacking exon 21 of Shank3, similar to PMS patients. Moreover, the RNAseq data have shown the sleep deprivation-induced increase of differential gene expression between Shank3^ΔC^ and wild-types, including the downregulation of circadian transcription factors. Reduced wheel-running activity in constant darkness is also reported for Shank3^ΔC^ mice. The results are potentially interesting and important. To strengthen the main conclusions of this paper, there are a few concerns that need to be addressed.

Essential revisions:

1) It is important to verify the RNAseq-identified changes in circadian transcription factors (*Per3, Dec2, Hlf, Tef, Reverbα*) in Shank3^ΔC^ mice with qPCR experiments. Since "Most of the differences in gene expression following SD are not present in HC conditions", do these circadian transcription factors show differences between WT vs. Shank3^ΔC^ only following SD?

2) To better understand the potential mechanisms underlying transcriptional dysregulation by deletion of exon 21 of Shank3, the authors need to provide evidence showing that the C-term truncated Shank3 (Shank3^ΔC^) accumulates in the nucleus of neurons from Shank3^ΔC^ mice. If no such nuclear accumulation of Shank3^ΔC^ can be found, an alternative mechanism is the nuclear translocation of some Shank3-interacting protein, such as β-catenin, in Shank3^ΔC^ mice (Qin L, et al., 2018), which leads to transcriptional changes. This possibility needs to be incorporated.

3) Shank3^ΔC^ mice have been reported to have impaired motor coordination (Kouser et al., 2013, but not Duffney et al., 2015). Could the wheel-running activity reduction of Shank3^ΔC^ mice in constant darkness (DD) result from motor-coordination deficits? Is there any reduction of wheel-running activity during 5-week regular light-dark cycles (LD) for Shank3^ΔC^ mice? What is the connection between sleep difficulties in Shank3^ΔC^ mice and reduced wheel-running activity in DD?

4) Results suffer from the absence of main statistics in support of the data (e.g., F and p values are absent for describing the genotype differences in the time course of wakefulness and NREM sleep presented in Figure 2A and Figure 3—figure supplement 1; the manner by which data in Figure 2—source data 1 were compared is unclear, and F or t values should be reported; F values and degrees of freedom also missing for Table 2). Moreover, it is very unclear how the analyses of repeated measures were performed at multiple instances. For example, an adequate analysis of the time course of Figure 2A would be to use two genotypes by 24 hours ANOVAs, but it seems from the description of the results that different parts of the 24-hour period were analyzed in separated ANOVAs (similar comment applies to Figure 3—figure supplement 1). This requires clarification and correction if necessary. Also, the use of paired t-tests for comparing genotypes in variables related to wheel-running activity as indicated in the methods is inadequate.

[Editors' note: further revisions were requested prior to acceptance, as described below.]

Thank you for resubmitting your work entitled "Shank3 Modulates Sleep and Expression of Circadian Transcription Factors" for further consideration at *eLife*. Your revised article has been favorably evaluated by Huda Zoghbi (Senior Editor), a Reviewing Editor, and two reviewers.

The manuscript has been improved but there are some remaining issues that need to be addressed before acceptance, as outlined below:

Although the revised manuscript has addressed many of the review comments, reviewer #3 pointed out statistical issues that remain to be further addressed. We feel that it is important for these points to be fully addressed.

*Reviewer #3:*

This revised manuscript presenting the sleep phenotype, responses to sleep loss and locomotor activity rhythm in a Shank3 mouse model of neurodevelopmental disease has been modified by authors in aiming to address the previous comments of the reviewers and editor. As indicated before, the presented work brings novelty with regards to molecular determinants of sleep disturbances in psychiatric diseases, and has value in its multi-methodology approach. However, authors seem to have only partly responded to previous concerns and the improvement of the manuscript in comparison to its original version is thus modest (e.g., no qPCR validation using the same RNA material as the one used for RNA-seq, no data regarding the nuclear location of the truncated Shank3, unclear description and inadequate use of statistics).

I will solely highlight below the points following up on my previous comments for which I feel that adequate modifications were not implemented.

1) Repeated-measure ANOVAs for time courses of vigilance states have been repeated multiple times for the same datasets (i.e., 1 – 24 hours, then again for 1 – 12 hours and 13 – 24 hours, then a third time for 1 – 6 hours, 7 – 12 hours, 13 – 18 hours, 19 – 24 hours). Such design is inadequate because applying numerous (unnecessary) statistical tests increases the chance of getting false positives (i.e., increase Type I error rate). The authors need to perform only one repeated-measure ANOVA per vigilance state (per full 24-hour time course) and to apply post hoc comparisons of individual hours afterwards. Post hoc comparisons of individual hours will also make sure the authors are not picturing significant differences for hours without differences such as what can be seen at the moment for hour 24 in Figure 2A for instance (see also stars showing significant differences covering hours with very similar means in Figure 3—figure supplement 1).

2) Moreover, repeated-measure ANOVAs have not been corrected for repeated measures using an appropriate correction method (e.g., Greenhouse-Geisser or Huynh-Feldt corrections). This needs to be done at all instances for which repeated-measure ANOVAs were used (e.g., Figure 2, Figure 2—figure supplement 1).

3) Some analyses are performed inadequately with regards to the design such as the number of bouts and duration of bouts of Figure 2—figure supplement 1 and of Figure 3—figure supplement 2 for which t-tests were used for each interval instead of appropriate application of a repeated-measure ANOVA that would test all intervals simultaneously. This should be corrected.

4) Similarly, data of EEG spectral bands presented under Supplementary file 1 are inappropriately analysed with multiple independent t-tests. This should be modified for a more adequate (and combined) statistical test (e.g., Genotype [WT vs. Shank3deltaC] by Period [light vs. dark] by Condition [baseline vs. sleep deprivation] ANOVAs).

5) Finally, with regards to statistics, the fact that statistics are only presented in the Source data files is extremely inconvenient and is making findings of the main manuscript very unclear. The full statistics should be presented in figure legends or in the main text of the manuscript. Moreover, the main manuscript should also clearly indicate the design of the statistical tests that were used for every variable without referring to other publications.

6) In the revised manuscript, the authors defined "hour 1" as the first hour of the light period and "hour 13" as the first hour of the dark period. Nevertheless, this is not described in the manuscript, and is thus even more confusing than the previous version of the manuscript in which Zeitgeber time (which is well established) was not correctly used.

---

## [Author Response]

Essential revisions:

*1) It is important to verify the RNAseq-identified changes in circadian transcription factors (Per3, Dec2, Hlf, Tef, Reverbα) in* Shank3^ΔC^
*mice with qPCR experiments. Since "Most of the differences in gene expression following SD are not present in HC conditions", do these circadian transcription factors show differences between WT vs.* Shank3^ΔC^
*only following SD?*

In consultation and agreement with the editors, it was decided that qPCR validation of the RNAseq results is outside of the scope of the current manuscript. This would require repeating a large part of the original study to obtain fresh samples. If the concerns are based on whether our RNA-seq results are reproducible, we addressed this issue by evaluating the ability of our study to reproduce findings from previous studies of genome-wide gene expression following sleep deprivation in mice as detailed in the methods section. Briefly, we assembled a list of 579 positive control genes found to be differentially expressed (FDR <0.005) by 5 hours of sleep deprivation relative to home cage controls in WT mice in 4 independent studies using two different microarray platforms and two different brain regions (Additional file 2, Gerstner et al., 2016). We then evaluated the ability of our RNA-seq study to recover the positive controls as differentially expressed when performing the same comparison done in the previous studies (HC5 vs SD5 in WT). Our RNA-seq study was able to recover 78% of the positive controls while being able to detect many more differences in gene expression. This is a very high level of concordance considering these controls were obtained using a different technology and data was collected from experiments in 4 independent laboratories. Therefore, we are confident in the ability of our study to detect differences in gene expression that can be independently validated. The Materials and methods section has been updated to make the reproducibility evaluation more clear and Figure 4—figure supplement 1 was added to show the results of this evaluation.

In regards to whether the differences in gene expression of circadian transcription factors is present in HC conditions, as shown in Table 1 three of the five circadian transcripts differentially expressed following SD are also downregulated in baseline HC conditions. In HC, *Per3* (FC -0.27; FDR 0.008), *Hlf* (FC -0.14; FDR 0.03) and *Tef* (FC -0.19; FDR 0.02) are downregulated in Shank3^ΔC^ mice. Nr1d1 (*Reverb*α) and Bhlhe41(*Dec2)* are not differentially expressed in HC. Genes differentially expressed in the mutants under HC and SD, as well as functional annotation of those genes, are reported in Figure 4, Figure 4—source data 1, and summarized in Table 1.

*2) To better understand the potential mechanisms underlying transcriptional dysregulation by deletion of exon 21 of Shank3, the authors need to provide evidence showing that the C-term truncated Shank3 (Shank3-ΔC) accumulates in the nucleus of neurons from* Shank3^ΔC^
*mice. If no such nuclear accumulation of Shank3-ΔC can be found, an alternative mechanism is the nuclear translocation of some Shank3-interacting protein, such as β-catenin, in* Shank3^ΔC^
*mice (Qin L, et al., 2018), which leads to transcriptional changes. This possibility needs to be incorporated.*

We thank the reviewer for suggesting this possible alternative mechanism. We amended the discussion to include Wnt signaling as an additional potential transcriptional regulatory mechanism as follows: “SHANK3 also modulates Wnt-mediated transcriptional regulation by regulating internalization of Wnt receptor Frizzled-2 (Harris et al., 2016) and nuclear translocation of the Wnt ligand β-catenin (Qin et al., 2018). The Wnt pathway kinase GSK3β phosphorylates circadian transcription factors PER2 (Iitaka et al., 2005), CRY2 (Harada et al., 2005) and REVERBα (Yin et al., 2006), however this mechanism modulates circadian period length which is not altered in Shank3^ΔC^ mice.”

*3)* Shank3^ΔC^
*mice have been reported to have impaired motor coordination (Kouser et al., 2013, but not Duffney et al., 2015). Could the wheel-running activity reduction of* Shank3^ΔC^
*mice in constant darkness (DD) result from motor-coordination deficits? Is there any reduction of wheel-running activity during 5-week regular light-dark cycles (LD) for* Shank3^ΔC^
*mice? What is the connection between sleep difficulties in* Shank3^ΔC^
*mice and reduced wheel-running activity in DD?*

The reviewer is correct that impaired motor coordination has been reported in Shank3^ΔC^ mice. To address the possibility that activity levels in Shank3^ΔC^ mice change across extended periods of wheel running due to motor coordination problems, we submitted a new cohort of animals to 5-weeks of voluntary wheel-running under regular 12:12 LD cycles. The results are shown in Supplementary file 2 and Source data 1. Under constant LD conditions, activity levels in Shank3^ΔC^ mice do not decrease over time. Therefore, we conclude that the reduction in wheel running we observed in constant darkness (DD) conditions is not due to motor problems.

As to the connection between sleep problems and reduced DD wheel-running activity, this is a very interesting question that we currently do not have enough data to address. In the future, simultaneous recordings of sleep and circadian activity will help us better answer this question and we thank the reviewer for the suggestion.

4) Results suffer from the absence of main statistics in support of the data (e.g., F and p values are absent for describing the genotype differences in the time course of wakefulness and NREM sleep presented in Figure 2A and Figure 3—figure supplement 1; the manner by which data in Figure 2—source data 1 were compared is unclear, and F or t values should be reported; F values and degrees of freedom also missing for Table 2). Moreover, it is very unclear how the analyses of repeated measures were performed at multiple instances. For example, an adequate analysis of the time course of Figure 2A would be to use two genotypes by 24 hours ANOVAs, but it seems from the description of the results that different parts of the 24-hour period were analyzed in separated ANOVAs (similar comment applies to Figure 3—figure supplement 1). This requires clarification and correction if necessary. Also, the use of paired t-tests for comparing genotypes in variables related to wheel-running activity as indicated in the methods is inadequate.

To better support the statistical analysis of our data, we added additional information to support the statistics for mouse EEG data as suggested by the reviewer, including F and t-values and degrees of freedom (Figure 2—source data 2 and Figure 3—source data 2)

Repeated measures ANOVAs on sleep data in Figure 2 and Figure 3 were performed as previously described (Ingiosi et al., 2015; Ingiosi and Opp, 2016). We have added those references to the methods section. It is not uncommon when performing analysis of sleep data in mice to consider separately periods of the day in which the amount and quality of sleep are known to be different (e.g. light versus dark phase). Post-SD data is usually analyzed considering the rest of the light phase following SD, and then the subsequent dark-phase. We added additional comparisons across longer time periods as suggested by the reviewer (Figure 2—source data 2 and Figure 3—source data 2).

Last, we agree with the reviewer and have removed the individual t-tests from the wheel-running analysis. ANOVA statistics for LD:DD and LD:LD wheel running experiments can be found in Table 2—source data 1 and Supplementary file 2—source data 1.

[Editors' note: further revisions were requested prior to acceptance, as described below.]Reviewer #3:[…] 1) Repeated-measure ANOVAs for time courses of vigilance states have been repeated multiple times for the same datasets (i.e., 1 – 24 hours, then again for 1 – 12 hours and 13 – 24 hours, then a third time for 1 – 6 hours, 7 – 12 hours, 13 – 18 hours, 19 – 24 hours). Such design is inadequate because applying numerous (unnecessary) statistical tests increases the chance of getting false positives (i.e., increase Type I error rate). The authors need to perform only one repeated-measure ANOVA per vigilance state (per full 24-hour time course) and to apply post hoc comparisons of individual hours afterwards. Post hoc comparisons of individual hours will also make sure the authors are not picturing significant differences for hours without differences such as what can be seen at the moment for hour 24 in Figure 2A for instance (see also stars showing significant differences covering hours with very similar means in Figure 3—figure supplement 1).

We included a repeated measures ANOVA on baseline data across the full 24 h per vigilance state in our first revision. We now also include repeated measures ANOVAs across the 19 hour recovery phase after sleep deprivation for each vigilance state. We removed the 12 hour- and 6 hour-binned repeated measures ANOVAs as suggested and now report posthoc pairwise comparisons with Sidak correction for individual hours when there are main effects of genotype or significant interactions between genotype and time. To provide a simpler presentation of overall light-dark differences in sleep time, we now include a new Table 1 that reports total sleep time in light and dark periods under baseline conditions.

2) Moreover, repeated-measure ANOVAs have not been corrected for repeated measures using an appropriate correction method (e.g., Greenhouse-Geisser or Huynh-Feldt corrections). This needs to be done at all instances for which repeated-measure ANOVAs were used (e.g., Figure 2, Figure 2—figure supplement 1).

All repeated measures ANOVAs were tested for sphericity, and a Greenhouse-Geisser correction was applied when appropriate. We updated the Materials and methods section to include this detail.

3) Some analyses are performed inadequately with regards to the design such as the number of bouts and duration of bouts of Figure 2—figure supplement 1 and of Figure 3—figure supplement 2 for which t-tests were used for each interval instead of appropriate application of a repeated-measure ANOVA that would test all intervals simultaneously. This should be corrected.

The reviewer is correct that statistical comparisons in Figure 2—figure supplement 1 and Figure 3—figure supplement 2 could be improved. To better support the statistical analysis of our data, we conducted repeated measures ANOVAs across all time intervals simultaneously for number of bouts and duration of bouts for each vigilance state. Posthoc pairwise comparisons with Sidak correction were conducted when permissible.

4) Similarly, data of EEG spectral bands presented under Supplementary file 1 are inappropriately analysed with multiple independent t-tests. This should be modified for a more adequate (and combined) statistical test (e.g., Genotype [WT vs. Shank3deltaC] by Period [light vs. dark] by Condition [baseline vs. sleep deprivation] ANOVAs).

The table in Supplementary file 1 was confusing as presented, added little to the study, and is now removed for clarity. Our principle EEG findings are that Shank3^ΔC^ mice have altered EEGs, accumulate NREM delta power after sleep deprivation, and show an abnormal homeostatic enhancement of higher frequencies after sleep deprivation. This information can be found in Figure 2B, Figure 3A, and Figure 3B–C, respectively. Removal of the Supplementary file 1 table simplifies the presentation of this information.

5) Finally, with regards to statistics, the fact that statistics are only presented in the Source data files is extremely inconvenient and is making findings of the main manuscript very unclear. The full statistics should be presented in figure legends or in the main text of the manuscript. Moreover, the main manuscript should also clearly indicate the design of the statistical tests that were used for every variable without referring to other publications.

We now report F and t statistics and p-values within the main text where specific findings are stated. We also added the statistical tests used in the figure legends. Due to the large number of statistical results that we are reporting, we followed the guidelines provided by the journal with respect to source data files:

“Authors should provide information about data processing and analysis in their figure legends, including any statistical tests applied, with exact sample number, p values of tests, criteria for data inclusion or exclusion, and details of replicates. In some cases, it might be unwieldy to have this information in the legend of a figure, in which case the information can be provided in a source data file.”

Additionally, we expanded the Materials and methods section to provide a clearer and more detailed description of the design of our statistical tests.

6) In the revised manuscript, the authors defined "hour 1" as the first hour of the light period and "hour 13" as the first hour of the dark period. Nevertheless, this is not described in the manuscript, and is thus even more confusing than the previous version of the manuscript in which Zeitgeber time (which is well established) was not correctly used.

All data are now presented as hours 1 – 24 which is also used in the sleep field and accessible to others outside of sleep and circadian research (who may be unfamiliar with the term ‘Zeitgeber’). We further clarified “hour 1” as light onset and “hour 13” as dark onset in the Materials and methods section. Changing to Zeitgeber time would require substantial edits to the text, figures, and source files, and this change would not impact the outcome of the experiments.